

# D-DI/PLT can be a prognostic indicator for sepsis

Xiaojun Zhao[1,*], Xiuhua Wu[2,*], Yi Si[1], Jiangang Xie[1], Linxiao Wang[1,3], Shanshou Liu[1], Chujun Duan[1], Qianmei Wang[1], Dan Wu[1], Yifan Wang[1], Jijun Chen[1], Jing Yang[1], Shanbo Hu[1], Wen Yin[1] and Junjie Li[1]

[1] Department of Emergency, Xijing Hospital, Fourth Military Medical University, Xian, Shaanxi, China
[2] Department of Respiratory and Clinical Care Medicine, Shanghai Sixth People's Hospital, Shanghai, China
[3] College of Life Sciences, Northwest University, Xi'an, Shaanxi, China
* These authors contributed equally to this work.

## ABSTRACT

**Aims:** To investigate the indicators affecting the early outcome of patients with sepsis and to explore its prognostic efficacy for sepsis.

**Methods:** We collected clinical data from 201 patients with sepsis admitted to the emergency department of Xijing Hospital between June 2019 and June 2022. The patients were categorized into groups (survival or fatality) based on their 28-day prognosis. The clinical characteristics, biochemical indexes, organ function-related indicators, and disease scores of the patients were analyzed for both groups. Risk factor analysis was conducted for the indicators with significant differences.

**Results:** Among the indicators with significant differences between the deceased and survival groups, D-dimer (D-DI), Sequential Organ Failure Assessment (SOFA) score, platelet (PLT), international normalized ratio (INR), and D-DI/PLT were identified as independent risk factors affecting the prognosis of sepsis patients. Receiver operating characteristic (ROC) curves showed that D-DI/PLT (area under the curve (AUC) = 93.9), D-DI (AUC = 89.6), PLT (AUC = 81.3), and SOFA (AUC = 78.4) had good judgment efficacy. Further, Kaplan Meier (K-M) survival analysis indicated that the 28-day survival rates of sepsis patients were significantly decreased when they had high levels of D-DI/PLT, D-DI, and SOFA as well as low PLTs. The hazard ratio (HR) of D-DI/PLT between the two groups was the largest (HR = 16.19).

**Conclusions:** D-DI/PLT may be an independent risk factor for poor prognosis in sepsis as well as a clinical predictor of patient prognosis.

# INTRODUCTION

Sepsis is a life-threatening disease with a high mortality rate (*Rudd et al., 2020*). Strategies have been implemented in recent years for sepsis early goal-directed therapies (EGDT) (*Huang et al., 2022*; *Evans et al., 2021*), such as the early identification of sepsis, improved diagnostic procedures, and timely antibiotic treatment, resulting in a reduction of overall

Corresponding authors
Wen Yin, xjyyyw@fmmu.edu.cn
Junjie Li, lijunjie@fmmu.edu.cn

mortality (*Khwannimit, Bhurayanontachai & Vattanavanit, 2019*; *Lien et al., 2022*; *Zhao et al., 2020a*). However, the under-diagnosis and delayed diagnosis of sepsis may also occur during pre-hospital care or emergency management. The introduction of "quick sequential organ failure assessment" (qSOFA) scores has improved clinical outcomes significantly; however, severe sepsis-induced death still accounts for two-thirds of hospital deaths, with a mortality rate of 17% (*Evans et al., 2021*). Sepsis is characterized by complex pathophysiological changes and involves multiple organ systems, resulting in less specific clinical signs at the early stage. Sepsis at the late stage can be diagnosed easily, but diagnosis at this stage misses the optimal treatment window, leading to increased mortality (*Pruinelli et al., 2018*). Therefore, the timely identification and early administration of empirical antibiotic regimens are required to improve the prognosis of septic patients and reduce their mortality rate (*Singer et al., 2016*).

The key to improving the treatment of sepsis lies in the ability to triage and treat it quickly when it is first identified, which is often the emergency department. Therefore, an evaluation index that can be adapted to the emergency department environment and provide a rapid indicator of the short-term prognosis of sepsis patients is needed. The National Early Warning Score (NEWS) and the Sequential Organ Failure Assessment (SOFA) were previously used to assess the prognosis of sepsis. However, the complex calculations and large number of tests required make them ineffective in predicting the prognosis and complications of sepsis (*Khwannimit, Bhurayanontachai & Vattanavanit, 2019*; *Oduncu, Kiyan & Yalcinli, 2021*). qSOFA can be used as a tool for the identification and prognosis assessment of sepsis patients, but its sensitivity and accuracy remain poor (*Goulden et al., 2018*; *Usman, Usman & Ward, 2019*). Therefore, an increasing number clinical studies have been carried out to explore the practical indicators for the rapid and accurate prediction of a sepsis prognosis (*Jiang et al., 2023*; *Arora, Mendelson & Fox-Robichaud, 2023*). The complete blood count (CBC) is a convenient test that has garnered recent interest for its use in the quick evaluation of the prognosis of sepsis (*Lien et al., 2022*; *Zhao et al., 2020a*). CBC parameters and relative parameter ratios, such as lymphocyte count, neutrophil count, platelets, neutrophil-to-lymphocyte ratio, platelets-to-lymphocyte ratio, *etc.*, have been widely used to predict the outcomes of the patients with preterm birth (*Zhang et al., 2022*) and COVID-19 survival rates (*Palladino, 2021*), as well as for hospital mortality (*Shimoni, Froom & Benbassat, 2022*). In the current retrospective study, we aimed to compare the clinical characteristics and biochemical indictors from initial routine examinations of sepsis patients that were admitted to the emergency department. These evaluations were used to assess the sensitivity of the indictors and the usefulness of the relative ratios in the prognosis of sepsis. We then sought to identify the appropriate prognostic indicators suitable for the sepsis in clinic.

## MATERIALS AND METHODS

### Patients

The clinical data of 389 patients with sepsis who were admitted to the emergency department of Xijing Hospital from June 1, 2019 to June 30, 2022 were retrospectively collected. Based on the inclusion and exclusion criteria, 188 patients who did not meet the
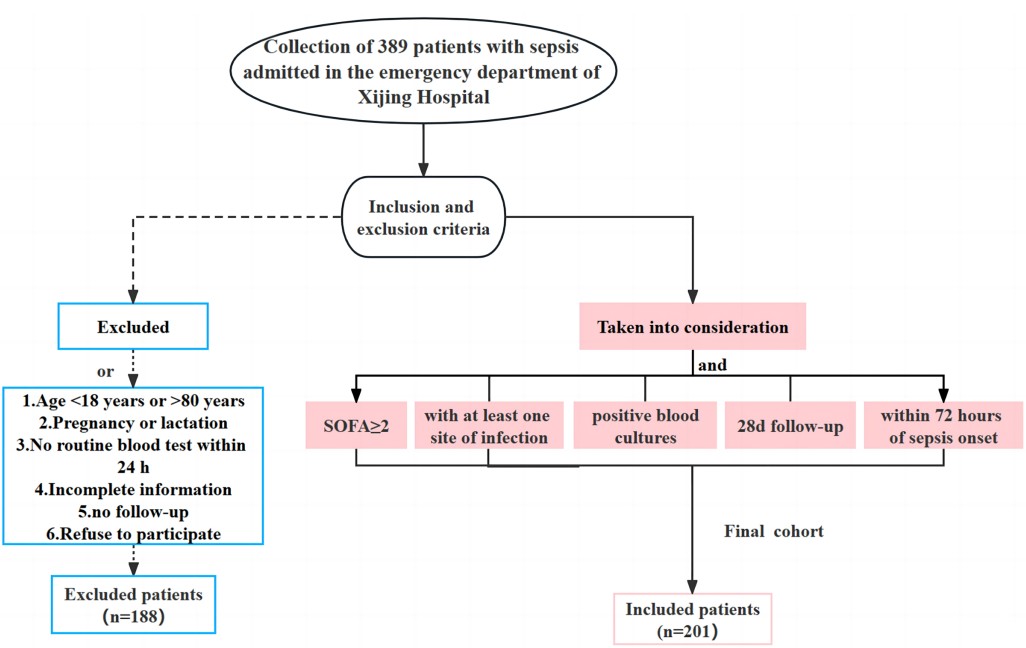

**Figure 1** **Flow chart of the clinical data collection for the patients with sepsis in the current study.**

requirements were excluded and a total of 201 subjects were retained. The inclusion criteria were: adult patients with a SOFA score of no less than two and at least one associated infection site within 24 h of admission; positive blood culture results; follow up for ≥28 days; sepsis onset within 72 h. The exclusion criteria were: age <18 years old or >80 years old; pregnancy or lactation; incomplete clinical data for the initial diagnosis; patients lost to follow-up or who declined to participate in the study (Fig. 1). Patients with sepsis were divided into survival and deceased groups based on their prognosis within 28 days. This study was approved by the ethics committee of our hospital (KY20212172-C-1), and the written informed consent was obtained from the patients and their families.

The clinical sample size was at least 98 cases. It was pre-calculated based on the morbidity and mortality rates of patients with sepsis that were admitted to our department, combined with the prognostic indicators of sepsis as exposure factors, the exposure rates of survival and death groups, the odds ratio or hazard ratio, the allowable errors with type I and type II errors, and a 10% rate of follow-up ($\alpha = 0.05$, $\beta = 0.2$, $1 - \beta$ as weights, $\Delta = 5\%$) (*Jia & Lynn, 2015*; *Wang & Ji, 2020*).

## Clinical data collection

The clinical data of all enrolled sepsis patients were collected and analyzed. This data included general clinical characteristics (including age, gender, hospital stay, basic/chronic diseases, comorbidities, ICU stay, pathogen infection, *etc.*), biochemical indicators and ratios, organ function related indicators (including pulmonary function, circulation function, kidney function, and liver function), and critical illness scores. More detailed information and significant indicators are described in the results section.

## Statistical analysis

A Kolmogorov-Smirnov test was conducted to determine the normal distribution of the continuous variables. Data with a normal distribution were presented as mean ± standard deviation and the comparisons between groups were analyzed by independent sample $t$ tests. Data with a non-normal distribution were expressed as medians (25–75% interquartile) and analyzed by non-parametric tests. Percentages (%) were used to express count data, and Chi-square or Fisher's exact tests were used to analyze the differences between groups. Further, Cox regression analyses were carried out to validate statistically significant indicators between groups and to screen for independent risk factors influencing the sepsis prognosis. Prognostic correlation heatmaps were used to analyze the relationship between risk factors derived from Cox regression analyses and prognosis. The receiver operating characteristic (ROC) curves were plotted to compare indicator potency through the area under the curve (AUC). Moreover, Kaplan Meier (K-M) survival analysis was used to verify the results from ROC. A $P$ value less than 0.05 was considered to be statistically significant.

# RESULTS

## Clinical characteristics and biochemical indicators of patients with sepsis between survival and fatality groups

A total of 201 subjects were included in the current study. These were divided into survival (141 patients) and fatality (60 patients) groups. The 7-day and 28-day mortality rates for each group were 14.4% and 29.85%, respectively, which is consistent with the range of sepsis mortality reported at home and abroad (*Rudd et al., 2020*; *Weng et al., 2023*; *Liu et al., 2023*). The age, temperature, respiratory rate, heart rate, diastolic blood pressure (DBP), systolic blood pressure (SBP) and other physiological indicators between the two groups were compared. We found that the average systolic blood pressure (115.80 ± 32.31 mmHg) and diastolic blood pressure (65.50 (35.00, 118.00) mmHg) in the fatality group were significantly lower than in the survival group. However, no significant difference was found in the age, temperature, respiratory rate, and heart rate between the two groups (Table 1). We compared the clinical features between the two groups and found that the survival group had an average length of stay of 7.89 ± 6.09 days. In this group there were 93 males (46.27%) and 48 females (23.88%); of these, 66 cases (32.84%) complicated with hypertension, 39 cases (19.40%) with diabetes, and 44 cases (21.89%) were admitted to ICU. In the fatality group, the average length of stay was 8.87 ± 6.54 days and the average age was 60.73 ± 17.40 years old. This group included 44 males (21.89%) and 16 females (7.96%). A total of 23 patients (11.44%) had cases complicated by hypertension, 13 patients (6.47%) had diabetes, and 27 patients were admitted to ICU. In addition, the patients' blood or body fluid samples were tested for pathogenic microorganisms. A total of 112 cases (55.72%) were infected with Gram-negative bacteria and 76 cases (37.81%) had Gram-positive bacterial infections. There were a small number of patients with double infections with Gram-positive and Gram-negative bacteria (five cases) and fungal infection (eight cases). The most common complication was acute liver injury (ALI) (70 cases,

**Table 1 General characteristics of sepsis patients.**

| Basic information | | Survival ($n$ = 141) | Death ($n$ = 60) | $P$ |
|---|---|---|---|---|
| Age (year) | | 57.82 ± 14.51 | 60.73 ± 17.40 | 0.26 |
| Temperature (°C) | | 37.05 ± 0.94 | 37.19 ± 1.08 | 0.38 |
| Respiratory rate (beats/min) | | 25.35 ± 7.20 | 26.40 ± 7.76 | 0.37 |
| Heart rate (beats/min) | | 102.04 ± 22.24 | 104.03 ± 28.45 | 0.63 |
| SBP (mmHg) | | 133.52 ± 27.12 | 115.80 ± 32.31 | **0.00** |
| DBP (mmHg) | | 79.00 (39.00, 165.00) | 65.50 (35.00, 118.00) | **0.00** |
| Hospital stay (day) | | 7.89 ± 6.09 | 8.87 ± 6.54 | 0.33 |
| Gender ($n$, %) | Female | 48 (23.88) | 16 (7.96) | 0.39 |
| | Male | 93 (46.27) | 44 (21.89) | |
| Hypertension ($n$, %) | No | 75 (37.31%) | 37 (18.41%) | 0.34 |
| | Yes | 66 (32.84%) | 23 (11.44%) | |
| Diabetes ($n$, %) | No | 102 (50.75%) | 47 (23.38%) | 0.48 |
| | Yes | 39 (19.40%) | 13 (6.47%) | |
| ICU | No | 97 (48.26%) | 33 (16.42%) | 0.09 |
| | Yes | 44 (21.89%) | 27 (13.43%) | |
| ALI ($n$, %) | No | 97 (48.26%) | 34 (16.92%) | 0.14 |
| | Yes | 44 (21.89%) | 26 (12.94%) | |
| AHI ($n$, %) | No | 118 (58.71%) | 43 (21.39%) | 0.08 |
| | Yes | 23 (11.44%) | 17 (8.46%) | |
| AKI ($n$, %) | No | 106 (52.74%) | 34 (16.92%) | **0.01** |
| | Yes | 35 (17.41%) | 26 (12.94%) | |
| ARF ($n$, %) | No | 104 (51.74%) | 32 (15.92%) | **0.01** |
| | Yes | 37 (18.41%) | 28 (13.93%) | |
| Pathogene | $G^+$ | 52 (25.87%) | 24 (11.94%) | 0.99 |
| | $G^-$ | 79 (39.30%) | 33 (16.42%) | |
| | $G^+ + G^-$ | 4 (1.99%) | 1 (0.50%) | |
| | Fungus | 6 (2.99%) | 2 (1.00%) | |

Note:
SBP, systolic blood pressure; DBP, diastolic blood pressure; ICU, intensive care unit; ALI, acute liver injury; AHI, acute heart injury; AKI, acute kidney injury; ARF, acute respiratory failure; $G^+$, gram positive bacteria; $G^-$, gram negative bacteria. Values with a $P$ < 0.05 are in bold.

34.83%), followed by acute respiratory failure (ARF) (65 cases, 32.34%) and acute kidney injury (AKI) (61 cases, 30.35%). Notably, complications with AKI and ARF in sepsis patients were significantly related to the prognosis. There were relatively few patients complicated with acute heart injury (AHI) (40 cases, 19.90%).

For biochemical indicators and related ratios, we found that the values of D-dimer-to-platelet ratio (D-DI/PLT), red blood cell distribution width-to-platelet ratio (RPR), neutrophil-to-lymphocyte ratio (NLR), platelet-to-lymphocyte ratio (PLR), D-DI (D-dimer), platelet (PLT), prothrombin time (PT), and international normalized ratio (INR) were significantly altered in the two groups (Fig. 2A). Organ function-related factors including circulation function-related brain natriuretic peptide (BNP) and creatine kinase

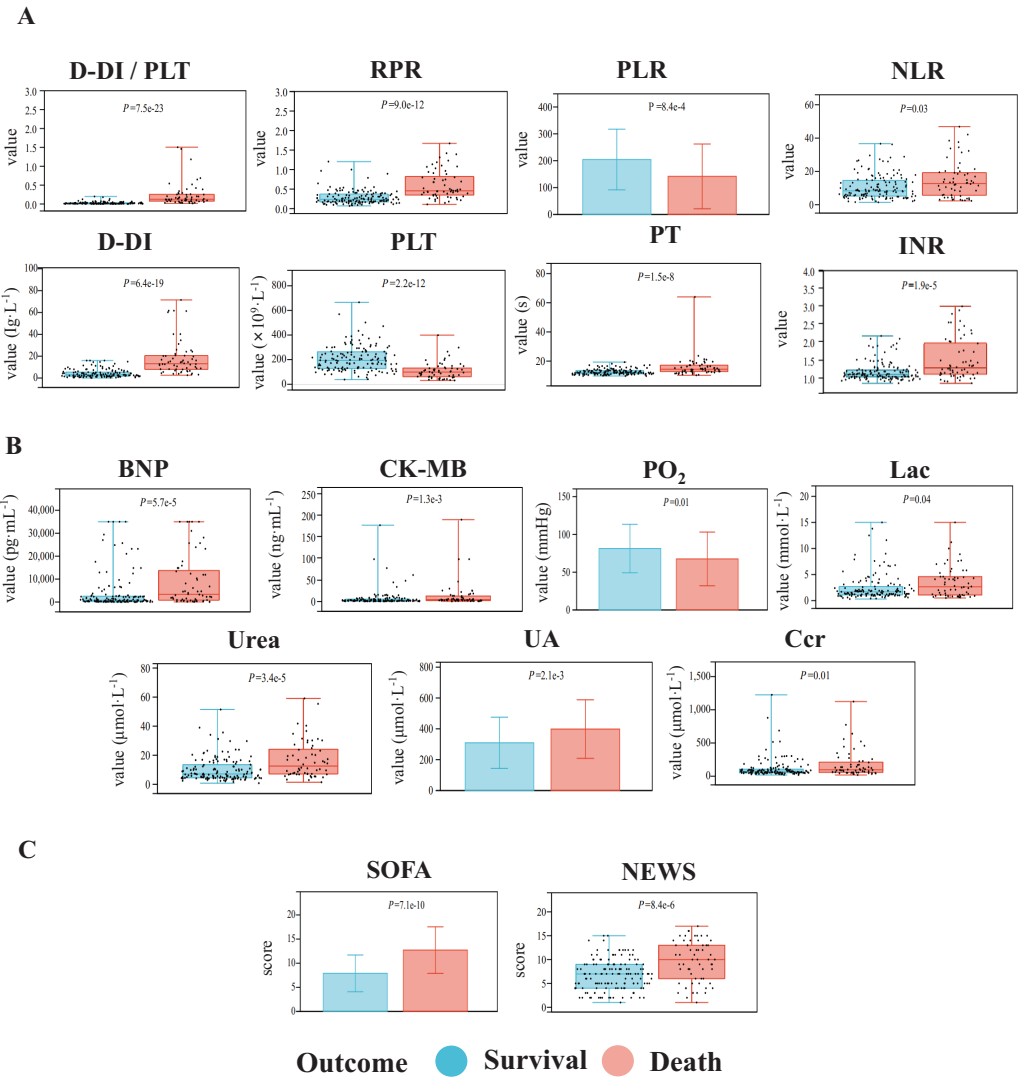

**Figure 2 Comparisons of the biochemical indexes (A), organ function-related indicators (B), and critical illness scores (C) of the patients with sepsis between the death group and the survival group.** The indicators with significant differences between the two groups are shown. The data with normal distribution are represented by bar charts, and the data with non-normal distribution are represented by box plots.

isoenzyme MB (CK-MB), pulmonary function-related partial pressure of oxygen (PO₂), liver function-related lactic acid (Lac) and kidney function-related urea, uric acid (UA), and creatinine clearance (Ccr), also showed significant differences between the two groups (Fig. 2B). Meanwhile, the SOFA and NEWS scores in the fatality group were significantly increased over those in the survival group (Fig. 2C). Patients with sepsis had no significant differences in the other clinical indicators between the two groups ($P \geq 0.05$, Table S1).

## Independent risk factors affect the prognosis of sepsis
Next, a multivariate Cox regression analysis was performed on the above-mentioned biochemical indicators with statistical significances between the two groups. As shown in

the forest plot, the hazard ratio (HR) of D-DI was 1.09 and the 95% confidence interval (95% CI) was [1.05–1.14]; the HR of INR was 3.28 and the 95% CI was [1.86–5.77]; the HR of SOFA was 1.20 and the 95% CI was [1.08–1.34]; the HR of PLT was 0.99 with a 95% CI of [0.98–1.00]; and the HR of D-DI/PLT was 4.06 with a 95% CI of [2.06–7.61]. All of these values had statistical significances, but the $P$ values for the remaining indicators were greater than or equal to 0.05 (Fig. 3A, Table S3). The results suggested that D-DI, SOFA, PLT, D-DI/PLT and INR are the independent risk factors that affected patient prognosis. We then used the predict function to calculate a multifactor risk score and plotted prognostic correlation heatmaps for these five indicators. The results showed that low PLT and high levels of D-DI/PLT, D-DI, SOFA and INR were associated with poor prognosis and high risk of death in sepsis patients (Fig. 3B).

### Efficacy of the screened prognostic indicators of sepsis

We examined the prognostic efficacy of D-DI, SOFA, PLT, D-DI/PLT, and INR based on their AUCs. According to the results, D-DI/PLT (AUC = 93.9, $P < 0.01$), D-DI (AUC = 89.6, $P < 0.01$), PLT (AUC = 81.3, $P < 0.01$), and SOFA (AUC = 78.4, $P < 0.01$) demonstrated good judgment efficacy, but INR was a weak indicator of the sepsis prognosis (AUC = 69.1) (Fig. 4A). We then excluded the INR index with an AUC less than 70% and drew K-M survival curves of the other four indicators based on their cut-off values of ROC to compare their relationship with the 28-day survival rate of patients with sepsis. It was shown that the 28-day survival rates were significantly decreased with high levels of D-DI/PLT, D-DI, SOFA and low levels of PLT, and vice versa (Fig. 4B). Notably, the hazard ratio (HR) between the two groups of D-DI/PLT was the greatest (HR = 16.19). D-DI/PLT is the combination of the values of D-DI and PLT and may be as a prognostic indicator of sepsis patients.

We further analyzed the potential predictive value of D-DI/PLT in the age, gender, disease state (hypertension, diabetes), and pathogen infection of sepsis patients. However, the results showed no significance between D-DI/PLT and these characteristics (Fig. S1).

### DISCUSSION

In the current study, we analyzed the clinical data of 201 patients with sepsis admitted to our department over the past three years to find effective prognostic indicators for sepsis. We found that 45% of the patients in the fatality group required ICU admission to support organ function. There was also a significantly higher mortality rate in this group due to greater incidences of respiratory failure or acute kidney injury compared with those in the survival group. This suggests that despite the development of several new diagnostic tests and treatment protocols for sepsis, patients with organ damage are often not seen in the prime diagnostic window and subsequently have poor overall outcomes. The high mortality rate of sepsis has not been fundamentally addressed and remains a major challenge for emergency and critical care physicians. Therefore, real-world clinical studies are needed to identify risk factors associated with poor prognosis in the early stages of sepsis. These studies could help clinicians quickly identify critically ill patients, provide timely intensification of treatment, and subsequently improve patient prognosis. We

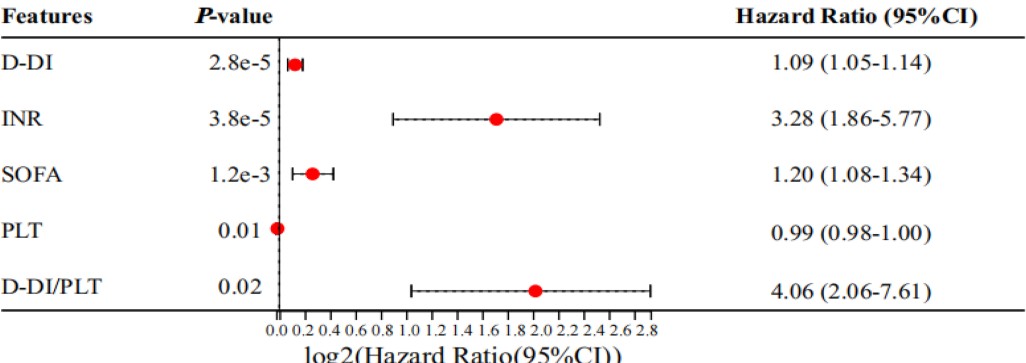

**A**

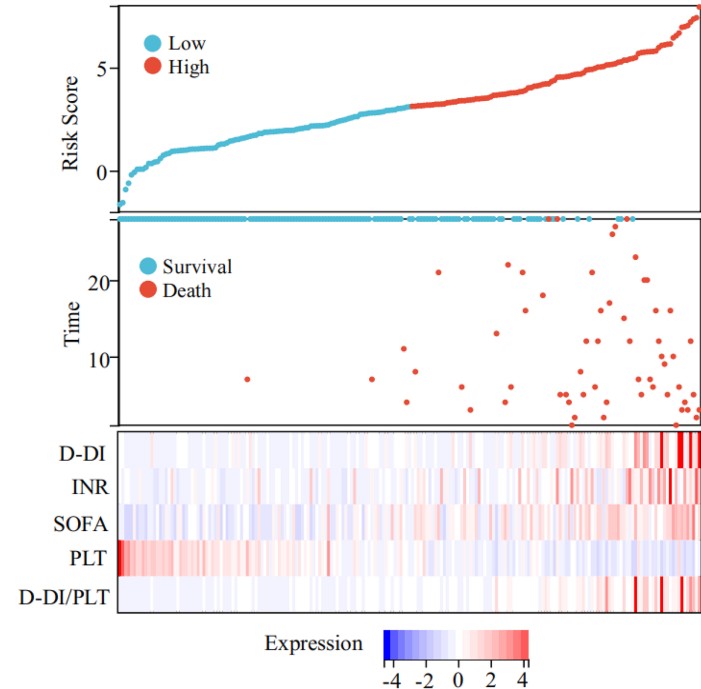

**B**

**Figure 3 Risk factors related to the prognosis of sepsis.** (A) Forest plot of Cox's stepwise regression results of the indicators with significant differences between the two groups. The forest plots show obvious differences in D-DI, INR, SOFA, PLT, and D-DI/PLT values between death and survival groups. Red points with dotted lines indicate the corresponding hazard ratio (HR) values with 95% confidence interval (CI). Factors with HR > 1 are risky, HR < 1 are protective, and HR = 1 have no significant difference. (B) The prognosis correlation heatmap of D-DI, INR, SOFA, PLT, and D-DI/PLT values. The top figure represents the patients' risk score, the middle scatter plot represents the patients' survival state, and the below heatmap shows the normalized expression values of the Z-scores for the above five factors.

analyzed the basic physiological and clinical biochemical indictors and found that the indexes of DBP, SBP, D-DI/ PLT, RPR, PLR, NLR, D-DI, PLT, PT, INR, urea, UA, Ccr, BNP, CK-MB, PO$_2$, Lac, SOFA, and NEWS showed significant differences in the sepsis

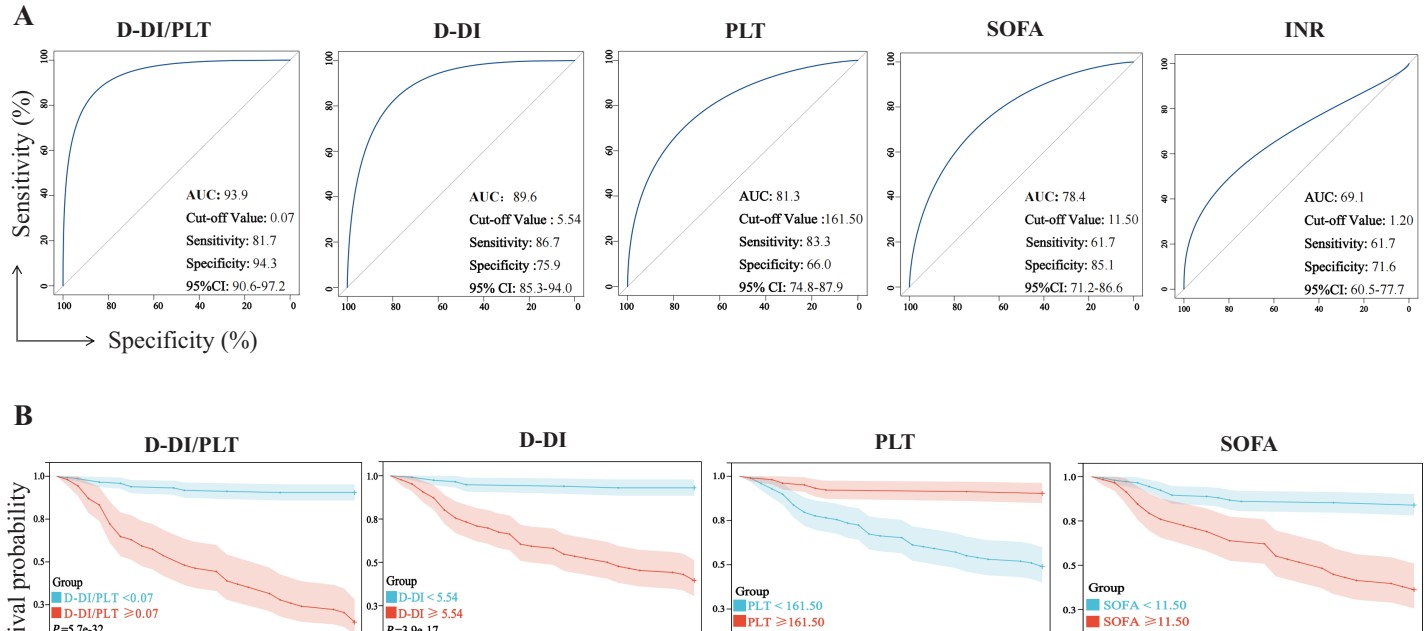

**A**

**Figure 4 Efficacy of the D-DI, INR, SOFA, PLT, and D-DI/PLT values in predicting the prognosis of sepsis.** (A) ROC curves of the D-DI, INR, SOFA, PLT, and D-DI/PLT values. The blue lines represent the metrics of analysis and the gray lines represent the reference lines. (B) K-M survival curves of the D-DI, SOFA, PLT, and D-DI/PLT values.            

patients between the survival and fatality groups. A Cox regression analysis was performed on the statistically significant indicators between the two groups, indicating that D-DI, SOFA, PLT, D-DI/PLT, and INR were independent risk factors influencing patient survival. We evaluated their prognostic efficacy using ROC curves and found that the AUCs of D-DI, SOFA, PLT, and D-DI/PLT were all greater than 0.70, and the predictive value of D-DI/PLT was the highest compared to other indicators. Further survival analysis showed that D-DI/PLT, D-DI, PLT, and SOFA were significantly correlated with the survival status of sepsis patients, and D-DI/PLT with the highest HR was the best predictor for 28-day survival of the patients.

D-DI/PLT is a combination of two clinical parameters, D-DI and PLT, which are readily accessible by routine blood testing. It is currently understood that the pathogenesis of sepsis is mainly due to an immune response imbalance and coagulation dysfunction (*Bachler et al., 2019*). Studies have shown that septic patients are affected by coagulation abnormalities, with approximately 50–70% of them developing clinically relevant coagulation dysfunction. Of these, 35% eventually progress to diffuse intravascular coagulation (*Levi & van der Poll, 2017*). As a biomarker of coagulation-fibrinolysis *in vivo*, D-DI can reflect the changes of coagulation in the body. These markers can be used in the assessment of all-cause mortality and guide the therapeutic measures for inflammatory vascular states, thrombotic events, and diseases associated with coagulation abnormalities

(*Favresse et al., 2018*; *Zhang et al., 2020*). Moreover, increasing evidence indicates that D-DI may be an independent prognostic factor for 28-day mortality and early clinical deterioration in septic patients (*Han et al., 2021*; *Lyngholm et al., 2019*). PLT is a nucleated cell that can cause bleeding disorders when its quantity and quality are altered. It plays an important role in promoting hemostasis and accelerating coagulation (*Hvas, 2016*). PLT also affects inflammatory and immune responses (*Koupenova et al., 2018*; *Mandel et al., 2022*). PLTs interact with immune cells by secreting and releasing pro-inflammatory factors and immune adhesion mediators, aggravating the inflammatory response and forming inflammatory thrombi (*de Bont, Boelens & Pruijn, 2019*; *Uzun et al., 2022*). During sepsis, the coagulation cascade and inflammatory response continuously cause PLT activation, even leading to DIC. Activated PLTs, in turn, drive the recruitment of leukocytes, monocytes, and neutrophils, forming an immune defense. The reduction of platelet production, increase of platelet consumption, and destruction of immune-mediated platelets eventually lead to a significant reduction in PLT amounts (*Vincent, Yagushi & Pradier, 2002*). Clinical studies have shown that PLT dysfunction is a risk factor for the poor prognosis of sepsis, with an increased risk of death in the patients with reduced platelet counts (*Zhao et al., 2020b*; *Ghimire et al., 2021*). Similarly, our study showed that sepsis patients in the fatality group had increased D-DI and decreased PLT within 24 h of admission, compared with the survival group. These results agree with the previous findings.

D-DI/PLT is a relatively new ratio and was initially used to differentiate maternal coagulation during pregnancy. The study found that D-DI/PLT values of disease-free pregnant woman were significantly lower than those of gestational woman with hypertension; however, there was no significant difference between the values in the later stages of pregnancy. This may be due to the fact that the coagulation and fibrinolytic system is not activated in the early pregnancy, while both late pregnancy and gestational hypertension are in a state of hypercoagulability. Hence, D-DI/PLT can be used as a predictor of hypercoagulability (*Limonta, Intra & Brambilla, 2022*). Similarly, the systemic inflammatory response activates the coagulation system during sepsis, forming a coagulation-inflammation network that puts the body in a state of hypercoagulability. The coagulation disorder-induced DIC in sepsis occurs earlier than multi-organ failure takes place (*Scarlatescu, Juffermans & Thachil, 2019*; *Iba et al., 2021*). Our study is the first to use D-DI/PLT as a prognostic indicator of sepsis. We found that D-DI/PLT had a higher specificity and was more accurate than other indicators used to judge prognosis. Survival analysis showed that the 28-day survival rate of sepsis patients with a D-DI/PLT ≥ 0.07 was less than 30%, and their risk of death was 16.19 times higher than that of the patients with a D-DI/PLT < 0.07, which is more pronounced than other indicators. Consistent with previous studies, D-DI/PLT can reflect the body's clotting system and an elevated D-DI/PLT value can be used as a predictor of poor prognosis of sepsis.

The D-DI/PLT value has the following advantages as a prognostic indicator of sepsis. Firstly, previous studies have shown that PLT is crucial for the development of sepsis. Activated PLTs contribute to inflammation and thrombosis, and the excessive consumption of PLT may lead to cytokine dysregulation in the body, disrupting the

immune system and inducing disseminated intravascular coagulation, ultimately leading to organ failure (*Assinger et al., 2019*). D-DI is one of the most commonly used coagulation indicators, especially in the assessment of thrombosis and the diagnosis and monitoring of DIC in the clinic (*Favresse et al., 2018*). In contrast to D-DI and PLT, D-DI/PLT is able to comprehensively evaluate more changes from various aspects. Secondly, compared with the SOFA score, D-DI/PLT can be obtained quickly through routine blood tests and the coagulation function test, and calculating the value is simple. Determining the D-DI/PLT value is suitable for the rapidity of the emergency department, allowing clinicians to more easily identify the development of sepsis and provide targeted treatment accordingly (*Moreno et al., 2023*). Moreover, some inflammatory markers (*e.g.*, C-reactive protein, TNF-α, and ferritin) and coagulation tests (*e.g.*, thromboelastography (*Zhu et al., 2023*), sublingual microcirculation (*Lu et al., 2023*), and dielectric coagulation (*Pourang et al., 2022*)) may not be detected in some primary care hospitals in developing countries, whereas D-DI and PLT values can be readily obtained through routine blood tests and coagulation tests that are reliable, simple, and inexpensive. Finally, the AUC of D-DI/PLT ratio was higher than those of D-DI, PLT, and SOFA as a whole, and its specificity was excellent. The D-DI/PLT value changes early in sepsis and emergency physicians can use it to predict the patients' short-term prognoses and stratify the treatment accordingly. However, some limitations existed in our study. The sample size was not large enough and there was a lack of multi-center joint research. In addition, the age range of the subjects in this study was too broad, ranging from 18–80 years, and the results were more focused on the adult population. There was also a lack of validation from an external cohort. In the future, we will continue to collect clinical information from patients with sepsis to enlarge the sample size and perform additional validation tests. Relevant data from newborns or young children should also be collected to expand the scope of the research applications. In conclusion, the D-DI/PLT value is an important tool for emergency physicians to use in order to predict the prognosis of septic patients and to stratify their treatment. It can help clinicians identify high-risk populations early and mitigate sepsis-induced hospital mortality rates.

### Funding

This work was supported by the Key research and development program of Shaanxi Province (Grant Number 2021SF-014), and the Basic research program of Natural Science in Shaanxi Province (Grant Numbers 2021JM-246, 2023-JC-YB-686, 2021SF-081, and 2022JQ-852). The funders had no role in study design, data collection and analysis, decision to publish, or preparation of the manuscript.

### Grant Disclosures

The following grant information was disclosed by the authors:
Key Research and Development Program of Shaanxi Province: 2021SF-014.

Basic research program of Natural Science in Shaanxi Province: 2021JM-246, 2023-JC-YB-686, 2021SF-081, and 2022JQ-852.

## Competing Interests

The authors declare that they have no competing interests.

## Author Contributions

- Xiaojun Zhao conceived and designed the experiments, performed the experiments, analyzed the data, prepared figures and/or tables, authored or reviewed drafts of the article, and approved the final draft.
- Xiuhua Wu conceived and designed the experiments, performed the experiments, analyzed the data, prepared figures and/or tables, and approved the final draft.
- Yi Si conceived and designed the experiments, analyzed the data, prepared figures and/or tables, authored or reviewed drafts of the article, and approved the final draft.
- Jiangang Xie performed the experiments, prepared figures and/or tables, authored or reviewed drafts of the article, and approved the final draft.
- Linxiao Wang performed the experiments, analyzed the data, prepared figures and/or tables, and approved the final draft.
- Shanshou Liu performed the experiments, prepared figures and/or tables, funding acquirement, and approved the final draft.
- Chujun Duan analyzed the data, prepared figures and/or tables, authored or reviewed drafts of the article, and approved the final draft.
- Qianmei Wang analyzed the data, authored or reviewed drafts of the article, and approved the final draft.
- Dan Wu performed the experiments, prepared figures and/or tables, and approved the final draft.
- Yifan Wang performed the experiments, prepared figures and/or tables, and approved the final draft.
- Jijun Chen performed the experiments, authored or reviewed drafts of the article, funding acquirement, and approved the final draft.
- Jing Yang performed the experiments, authored or reviewed drafts of the article, funding acquirement, and approved the final draft.
- Shanbo Hu performed the experiments, authored or reviewed drafts of the article, funding acquirement, and approved the final draft.
- Wen Yin conceived and designed the experiments, authored or reviewed drafts of the article, and approved the final draft.
- Junjie Li conceived and designed the experiments, authored or reviewed drafts of the article, funding acquirement, and approved the final draft.

## Human Ethics

The following information was supplied relating to ethical approvals (*i.e.*, approving body and any reference numbers):

The Ethical Committee of the Xijing Hospital of Fourth Military Medical University.

## Data Availability

The raw measurements are available in the Supplemental Files.

## Supplemental Information

Supplemental information for this article can be found online at http://dx.doi.org/10.7717/peerj.15910#supplemental-information.

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
