# Peer review of "D-DI/PLT can be a prognostic indicator for sepsis"

_PeerJ, doi:10.7717/peerj.15910_

## Round 0.1 · original submission · Minor Revisions

The biomarkers of sepsis need some update as per literature published in recent years. The present study explores and provides a good biomolecule that can provide strong pavement in the same field.

Incorporating the suggestions of the reviewers will add strength to the manuscript. For example, one of the reviewer has asked to add information about the status of the culture positivity of all samples included in the study. The materials and methods contains huge clinical and biochemical data which was not explained/discussed in the manuscript. The authors should either provide justification for the same or remove it. Therefore the comments of all reviewers should be addressed.

The decision of minor revision can be justified as all three reviewers have suggested that the manuscript is well written and the experiments are nicely executed. The results can add new information in the prognosis of the disease. Therefore, improving the manuscript under ‘minor review’ will be sufficient to publish it.

Reviewer 1 ·

Basic reporting

The manuscript is generally clear and easy to understand. The English language could be further improved. The introduction could be further refined to emphasize the novelty and clinical value of the study. The article structure is ok. Raw data shared.

Experimental design

This is a retrospective study to identify the risk factors of mortality for sepsis patients. Sepsis patients were divided into the survivor and non-survivor groups. Clinical and laboratory data were collected and multivariate logistic regression analysis was performed to determine the independent risk factors for mortality. The predictive value of DD/PLT as shown by the ROC curve and AUC was superior to traditional SOFA score. As stated by the authors, this might be the first to use D-D/PLT as a prognostic indictor for sepsis. The findings are of clinical significance. There are some comments for the improvement of this study as follows:

1. To get a clear profile of the study cohort, the baseline demographic and clinical characteristics are suggested to be described in more details. For example, the age, gender, underlying diseases, the infection site and pathogens could be introduced.

2. The sepsis patients were divided into the death and survival group. Whether patients were categorized according to the 28-day survival or survival depending on the follow-up should be clarified. If it’s the latter, the mean follow-up should be described.

3. Since sepsis is a very heterogenous disease, t’s interesting to know the predictive value of DD/PLT in different subgroups.

Validity of the findings

It’s a retrospective study conducted in a single center. The sample size is relatively small. It’s better to further validate the findings with an external cohort.

Cite this review as

Reviewer 2 ·

Basic reporting

Choice of words/repetition should be taken cared of.
literature should be updated (for 2023)
Raw data should be supplied.
were there any comorbidities present in the patients enrolled?
The effect of comorbidities of the patients enrolled must be presented as these can directly affect the results.

Experimental design

Numerous data were collected (described in Methods: Clinical Data Collection), but the majority of them are neither presented in the results nor described in the discussion part of MS.
what was the motto behind the collection of huge data?
was there any correlation with the objective?
what were the results of those data?
what was the correlation wrt objective?

Validity of the findings

The inclusion of results of numerous data collected may alter the current findings which are otherwise acceptable.

Additional comments

the reference list should be updated by including articles from 2023.

Cite this review as

Reviewer 3 ·

Basic reporting

1. Author are requested to please go through the present proposal for the English language and grammatical errors. (eg Please see Discussion , line 170).
2. statistical data on sepsis, mortality may have been added.

Experimental design

1. The methods described in the present study was not clear as in the Introduction part, line number retrospective study was mentioned and not properly defined elsewhere.
2. culture positivity of specimen was not defined for sepsis as blood culture is the gold standard for sepsis.

Validity of the findings

robust and statistically sound, rigorous work done
should include analysis among different age groups.

1. Figure 1 should be elaborated
2. Figure 4 was not clear
3. no details of patient location, ICU critical care etc.

Additional comments

1. Above submitted Proposal is needed to be completely revised with full details along with appropriate introduction, methodology with detail workplan, and conclusion as very limited information’s are available in the current submission.
2. How the Sample size was calculated was not mentioned in the proposal.
3. Data for Additional factors such as different age groups, patient enrollment into the study, any patient went Left against Medical Advice (LAMA), septic patient defined criteria, any culture positive specimen, follow up of patient details may also needed to capture in order to add more impact for analysis in the study proposal.
4. Inclusion criteria and exlsuion criteria also needed to clarify in details whether patient denied consent or clinically unstable cases were enrolled in the study as the patient enrollment was not mentioned in the current study.
5. It’s a very interesting to study the D-DI/PLT as a prognostic indicator for sepsis in the neonates/young children requiring intensive care facility.

Cite this review as

---

## Round 0.2 · accepted · Accept

Based on the comments from both Reviewer 1 and Reviewer 2, it is evident that the authors have satisfactorily addressed all the comments and recommendations provided during the review process. The revised manuscript now meets the relevant criteria for acceptance, and the authors have updated the data input and structure as per the reviewers' suggestions. Additionally, all missing data have been added, and the revised manuscript justifies the analysis described.

Therefore, I recommend accepting the manuscript for publication in its present form.

Reviewer 1 ·

Basic reporting

The authors have addressed the comments satisfactorily.

Experimental design

The authors have addressed the comments satisfactorily.

Validity of the findings

The authors have addressed the comments satisfactorily.

Cite this review as

Reviewer 2 ·

Basic reporting

the revised manuscript incorporates all suggestions and recommendations. Now the revised manuscript passes relevant criteria to be accepted.

Experimental design

the authors have update the data input, structure as per recommendations.
the revised data accomplish objectives.

Validity of the findings

All missing data have been added in revised manuscript.
The revised manuscript justifies the analysis described.

Cite this review as